# Dipotassium Glycyrrhizinate on Melanoma Cell Line: Inhibition of Cerebral Metastases Formation by Targeting NF-kB Genes-Mediating MicroRNA-4443 and MicroRNA-3620—Dipotassium Glycyrrhizinate Effect on Melanoma [note 1]

**DOI:** 10.3390/ijms23137251

**Published:** 2022-06-29

**Authors:** Gabriel Alves Bonafé, Jéssica Silva dos Santos, Jussara Vaz Ziegler, Fernando Augusto Lima Marson, Thalita Rocha, Manoela Marques Ortega

**Affiliations:** 1Laboratory of Cell and Molecular Tumor Biology and Bioactive Compounds, Post Graduate Program in Health Science, São Francisco University (USF), Avenida São Francisco de Assis, 218, Bragança Paulista 12916-900, São Paulo, Brazil; gabrielbonafe@outlook.com (G.A.B.); santosjssica97@yahoo.com (J.S.d.S.); fernandolimamarson@hotmail.com (F.A.L.M.); 2Laboratory of Human and Medical Genetics, Post Graduate Program in Health Science, USF, Bragança Paulista 12916-900, São Paulo, Brazil; 3Verdi Cosmetics LLC, Joanópolis 12980-000, São Paulo, Brazil; jussaraziegler@verdicosmeticos.com.br; 4Postgraduate Program in Biomaterials and Regenerative Medicine, Faculty of Medical Sciences and Health, Pontifical Catholic University of São Paulo, Sorocaba 05014-901, São Paulo, Brazil; tharochaster@gmail.com

**Keywords:** Dipotassium Glycyrrhizinate, anti-migratory effect, melanoma cell line SK-MEL-28, NF-kB pathway inhibition, miR-4443 and miR-3620, *CD209*, *TCN* genes modulation

## Abstract

Glycyrrhizic acid (GA), a natural compound isolated from licorice (*Glycyrrhiza glabra*), has exhibited anti-inflammatory and anti-tumor effects in vitro. Dipotassium glycyrrhizinate (DPG), a dipotassium salt of GA, also has shown an anti-tumor effect on glioblastoma cell lines, U87MG and T98G. The study investigated the DPG effects in the melanoma cell line (SK-MEL-28). MTT assay demonstrated that the viability of the cells was significantly decreased in a time- and dose-dependent manner after DPG (IC_50_ = 36 mM; 24 h). DNA fragmentation suggested that DPG (IC_50_) induced cellular apoptosis, which was confirmed by a significant number of TUNEL-positive cells (*p*-value = 0.048) and by *PARP-1* [0.55 vs. 1.02 arbitrary units (AUs), *p*-value = 0.001], *BAX* (1.91 vs. 1.05 AUs, *p*-value = 0.09), and *BCL-2* (0.51 vs. 1.07 AUs, *p*-value = 0.0018) mRNA compared to control cells. The proliferation and wound-healing assays showed an anti-proliferative effect on DPG-IC_50_-treated cells, also indicating an inhibitory effect on cell migration (*p*-values < 0.001). Moreover, it was observed that DPG promoted a 100% reduction in melanospheres formation (*p*-value = 0.008). Our previous microRNAs (miRs) global analysis has revealed that DPG might increase miR-4443 and miR-3620 expression levels. Thus, qPCR showed that after DPG treatment, SK-MEL-28 cells presented significantly high *miR-4443* (1.77 vs. 1.04 AUs, *p*-value = 0.02) and *miR-3620* (2.30 vs. 1.00 AUs, *p*-value = 0.01) expression compared to control cells, which are predicted to target the *NF-kB*, *CD209* and *TNC* genes, respectively. Both genes are responsible for cell attachment and migration, and qPCR revealed significantly decreased *CD209* (1.01 vs. 0.54 AUs, *p*-value = 0.018) and *TNC* (1.00 vs. 0.31 AUs, *p*-value = 2.38 × 10^−6^) mRNA expression levels after DPG compared to untreated cells. Furthermore, the migration of SK-MEL-28 cells stimulated by 12-O-tetradecanoylphorbol-13-acetate (TPA) was attenuated by adding DPG by wound-healing assay (48 h: *p*-value = 0.004; 72 h: *p*-value = 7.0 × 10^−4^). In addition, the MMP-9 expression level was inhibited by DPG in melanoma cells stimulated by TPA and compared to TPA-treated cells (3.56 vs. 0.99 AUs, *p*-value = 0.0016) after 24 h of treatment. Our results suggested that DPG has an apoptotic, anti-proliferative, and anti-migratory effect on SK-MEL-28 cells. DPG was also able to inhibit cancer stem-like cells that may cause cerebral tumor formation.

## 1. Introduction

Glycyrrhizic acid (GA; C_42_H_62_O_16_), a natural compound isolated from licorice (*Glycyrrhiza glabra*), has exhibited anti-inflammatory and anti-tumor effects on human hepatoma, promyelocytic leukemia, stomach cancer, Kaposi sarcoma-associated herpes virus-infected cells, and prostate cancer cells in vitro [1,2,3]. Dipotassium glycyrrhizinate (DPG, C_42_H_60_K_2_O_16_), a dipotassium salt of GA, also has presented anti-inflammatory characteristics but without the side effects observed by direct GA administration. Recently, we have demonstrated that DPG presented an anti-tumor effect on glioblastoma cell lines, U87MG and T98G, through a decrease in proliferation and an increase in apoptosis. In addition, our data also suggested that the DPG anti-tumor effect is related to nuclear factor kappa B (NF-kB) pathway suppression, where interleukin-1 receptor-associated kinase 2 (*IRAK2*)- and TNF receptor-associated factor (*TRAF6*)-mediating microRNA (*miR*)*-16* and *miR-146a*, respectively, might be a potential therapeutic target of DPG [4].

Melanomas are the third most common cause of cerebral metastases, preceded only by non-small-cell lung cancer and breast carcinomas [5]; moreover, more than 50% of patients with metastatic melanoma have a specific mutation in the serine/threonine kinase *BRAF* [6]. The risk of brain metastases in metastatic melanoma increases with disease duration. Melanoma brain metastases have been identified in up to 75% of patients with metastasized melanoma [7,8]. Moreover, melanoma brain metastasis is associated with poor prognosis, and it has been reported that the median time from primary melanoma diagnosis to brain metastasis is around 3.2 years [9].

The spectrum of available treatments for metastatic melanoma has increased substantially over the last six years due to the approval of effective immunotherapies and targeted treatments. However, until recently, patients with brain metastases have been excluded from most clinical studies, and prognosis has remained poor, with survival typically measured in a few months if untreated [10].

Anti-apoptotic nuclear factor κB (NF-kB) is one important pathway that melanoma tumors use to achieve survival, proliferation, resistance to apoptosis, and metastasis. In fact, it has been demonstrated that upregulation of NF-kB levels is involved in both the progression of melanoma [11] and the increase of its metastatic potential [12]. Therefore, inhibition of NF-kB activation appears to be a promising option for anti-cancer therapies. In melanoma cells, a study has highlighted that some components of the NF-kB pathway family, such as p50 and p65 (RelA) subunits, are overexpressed in the nuclei of dysplastic nevi and melanoma cells compared to those of normal nevi and healthy melanocytes, respectively [13]. Besides, previously published work reported that melanocytes expressing a conditionally oncogenic form of the *BRAF*^V600E^ mutation exhibit enhanced β-Trcp expression, increased kappa β kinase (IKK) activity, and a concomitant increase in the rate of IκBα degradation. Conversely, inhibition of *BRAF* signaling using either a broad-spectrum Raf inhibitor or by selective knockdown of *BRAF*^V600E^ expression by RNA interference in human melanoma cells leads to decreased IKK activity and β-Trcp expression, stabilization of IκB, inhibition of NF-kB transcriptional activity and sensitization of these cells to apoptosis [14].

The present study investigated DPG effects in the melanoma cell line (SK-MEL-28) bearing *BRAF* mutation.

## 2. Material and Methods

### 2.1. Melanoma Cell Line and Reagents

The human melanoma cell line (SK-MEL-28) was donated by Dr. Gustavo Jacob Lourenço from the University of Campinas, São Paulo, Brazil, and it was cultured at Dulbecco’s modified Eagle’s medium (DMEM) high glucose supplemented with 10% fetal calf serum (FCS) and 1% streptomycin/penicillin (Cultilab, Campinas, São Paulo, Brazil) at 37 °C in a 5% CO_2_ atmosphere.

DPG [chemical abstracts service (CAS) number 68797-35-3] and 12-O-Tetradecanoylphorbol-13-acetate (TPA) (CAS number 16561-29-8) were obtained from Verdi Cosmetics LLC (Joanópolis, São Paulo, Brazil) and Sigma Chemical (St. Louis, MO, USA), respectively. For cell line treatments, DPG and TPA were diluted in DMEM to prepare a 2000 μM and 50 ng/μL stock solution, respectively. All treatment assays were performed in the presence of 10% FCS.

### 2.2. Determination of Cellular Metabolic Activity (Cell Viability)

Cell viability was determined by [(3-(4, 5-dimethyl thiazolyl-2)-2, 5-diphenyltetrazolium bromide)] (MTT) assay. Briefly, adherent melanoma cells (SK-MEL-28) and the non-tumoral keratinocyte cell line (HaCat) cells were seeded in 96-well flat-bottomed tissue culture plates at 0.2 × 10^6^ cells/plate in 100 µL DMEM containing 10% FBS and 1% penicillin/streptomycin and various concentrations (5 mM, 8 mM, 12 mM, 15 mM, 18 mM, 20 mM, 24 mM, 28 mM, 32 mM, 36 mM, and 40 mM) of DPG based on a previous study (Bonafé et al., 2019). The cells were cultured for 24 h, 48 h, and 72 h prior to treatment with 100 µL of 0.2 μg/μL of MTT working solution (Sigma, St. Louis, MO, USA) for 4 h at 37 °C. Following incubation, formazan crystals were solubilized with 100 µL of dimethyl sulfoxide (DMSO). Cell viability was determined by measuring the optical density at 550 nm using a microplate spectrometer (Thermo Fisher, Waltham, MA, USA). Cell survival rates were expressed as percentages of the value of normal cells. Untreated control cells were analyzed in all experiments, and all DPG dose treatments were performed in triplicate.

### 2.3. Measurement of the Effect of DPG on Cell Viability In Vitro

Melanoma cell line (SK-MEL-28) (0.4 × 10^6^ cells/well) was cultured in 24-well tissue culture plates in the presence of DPG (36 mM). Cells were washed in phosphate-buffered saline (PBS) solution, and viable cell count was determined by trypan blue dye exclusion assay for four days. Untreated cells were used as a control, and the experiments were performed in triplicate.

### 2.4. Agarose Gel Electrophoresis Analysis for DNA Fragmentation

SK-MEL-28 cells were cultured in 6-well tissue culture plates for 24 h at 37 °C in a 5% CO_2_ environment. Cell lines were then exposed to 36 mM of DPG for 24 and 48 h. At the end of the experimental period, cells were collected and suspended in 200 μL of PBS. The DNA was isolated using lithium chloride extraction [15]. The purity of DNA was analyzed in a spectrophotometer at 260/280 nm, and the ratio was confirmed to be between 1.7 and 1.9. DNA samples were then electrophoresed on a 1.5% agarose gel and visualized with ethidium bromide staining under UV illumination.

### 2.5. Apoptosis by TUNEL (Terminal Deoxynucleotidyl Transferase dUTP Nick End Labeling) Assay

Melanoma cells were cultured in 96-well tissue culture plates for 24 h. Cell lines were then exposed to 36 mM for 48 h. Apoptosis was evaluated on cells after treatment with DPG by in situ TUNEL assay, using the in-situ cell death detection kit, fluorescein (Roche Applied Science, Mannheim, Germany), according to the manufacturer’s protocols. Apoptotic indexes were calculated by scoring four randomly selected fields and counting the number of apoptotic cells over the total number of viable cells, representing a quota compared to untreated cells. Cells were directly analyzed under a fluorescence microscope (Axio Vert. A1 ZEISS, Germany).

### 2.6. Wound-Healing Assay

SK-MEL-28 cells were seeded in six-well plates and grown overnight at the confluence. The monolayer of cells was scratched with a 200 μL pipette tip to create a wound, and the plates were washed twice with PBS and cultured with DPG (36 mM), TPA (50 ng/mL), and both DPG + TPA (36 mM and 50 ng/mL, respectively). After cells migrating from the leading edge were photographed at 0, 24, 48, and 72 h under an inverted microscope (Axio Vert. A1 ZEISS). Untreated cells were used as a control. The distance of the scratch closure was examined using ImageJ software (National Institutes of Health, Bethesda, MD, United States). Each value is derived from the same selected fields, and the results are expressed as the mean of migrating cell numbers per field. Each treatment (DPG, TPA, and DPG + TPA) was examined in triplicate.

### 2.7. DPG Effect on MicroRNAs (miRs)

A global analysis of miRs (Affymetrix Human miRNA 4.0, Applied Biosystems™, Santa Clara, CA, United States of America), considering only miRs previously known as NF-kB pathway genes regulators, was previously performed by our research group using glioblastoma Temozolomide (TMZ) resistant T98G cells. The analysis of the differential expression profile of miRs was performed considering only 91 miRs previously selected and predicted as regulators of genes involved with the NF-kB pathway using miRanda (http://www.microrna.org, accessed on 4 October 2019), Targetscan (http://www.targetscan.org, accessed on 4 October 2019), and Findtar (http://bio.sz.tsinghua.edu.cn, accessed on 4 October 2019).

The global analysis revealed that DPG increased *miR-4443* and *miR-3620* expression levels C, which are predicted (99%) to target the NF-kB post-transcriptional genes, Cluster of Differentiation 209 (*CD209*) and Tenascin C (*TNC*), respectively. Thus, the present study evaluated the expression of both miRs and their predicted target genes, *CD209* and *TNC,* by quantitative polymerase chain reaction (qPCR) using SK-MEL-28 treated with DPG cells. Total RNA was isolated using Trizol (Invitrogen^®^, Carlsbad, California, USA). *MiR-4443* (assay 463010_mat), *miR-3620* (assay CTKA3MT), and *U6* (assay 001973) (assay 03928990_g1) cDNA were synthesized from total RNA according to the TaqMan^®^ qPCR assays protocol (Applied Biosystems^®^, Foster City, CA, USA)

The expression value of each gene was represented in arbitrary units (AUs) in triplicate samples (Appendix A).

### 2.8. Reverse-Transcriptase Polymerase Chain Reaction (RT-PCR) Analysis

Total RNA was isolated using Trizol^®^ reagent (Thermo Fisher Scientific, Waltham, MA, USA), according to the manufacturer’s instructions from SK-MEL-28 cells exposed to DPG (36 mM), TPA (50 ng/mL), and both DPG + TPA (36 mM and 50 ng/mL, respectively) for 48 h. The non-tumoral HaCat cells were also treated using DPG (18 mM) for 48 h. Untreated cells were used as a control.

Total RNA (1 µg) was converted to cDNA by High-Capacity cDNA Reverse Transcription Kit (Applied Biosystems, Foster City, CA, USA) according to the manufacturer’s protocol. Oligonucleotide primer sequences used were as follows: BCL2 Apoptosis Regulator (*BCL2*) 5′- GTGGATGACTGAGTACCTGAAC-3′ (sense) and 5′- GAGACAGCCAGGAGAAATCAA-3′(anti-sense); BCL2 Associated X (*BAX*) 5′- TTCTGACGGCAACTTCAACT-3′ (sense) and 5′- CAGCCCATGATGGTTCTGAT-3′(anti-sense); Poly [ADP-ribose] Polymerase 1 (*PARP-1*) 5′- GCCGAGATCATCAGGAAGTATG-3′ (sense) and 5′- ATTCGCCTTCACGCTCTATC-3′(anti-sense); *CD209* 5′- CATGTCTAACTCCCAGCGG-3′ (sense) and 5′- GAAAGTCCCATCCAGGTGAAG-3′(anti-sense); *TNC* 5′- CACTACACAGCCAAGATCCAG-3′ (sense) and 5′- TCGTGTCTCCATTCAGCATTG-3′(anti-sense); Matrix Metalloproteinase 9 (*MMP-9*) 5′-CACTGTCCACCCCTCAGAGC-3′ (sense) and 5′-GCCACTTGTCGGCGATAAGG-3′(anti-sense); and Glyceraldehyde-3-Phosphate Dehydrogenase (*GAPDH*) 5′- GCACCGTCAAGGCTGAGAAC-3′(sense) and 5′- CCACTTGATTTTGGAGGGAT-3′(antisense). Relative quantification of targets was determined by the delta delta cycle threshold (ΔΔCt) method [16]. Each sample was examined in triplicate and the raw data were presented as the relative quantity of the targets, normalized by *GAPDH* (Appendix A).

### 2.9. Sphere-Cultured Stem Cell-Enriched Melanoma Populations

Melanoma cells (10,000 cells/mL) were cultured at 37 °C in a 5% CO_2_ atmosphere in serum-free DMEM/F12 supplemented with N_2_ supplement (StemCell, Vancouver, Canada) containing epidermal growth factor (EFG) (20 ng/mL) and basic fibroblast growth factor (20 ng/mL) (Peprotech, Ribeirão Preto, São Paulo, Brazil) and 1% penicillin (120 mg/mL)/streptomycin (120 mg/mL; Thermo Fisher Scientific, Inc.) for at least 6 days. For subsequent DPG treatment (36 mM), 75-μm neuro-spheres were cultured 24, 48, and 72 h. Cells without DPG were cultured as controls. Cells were observed and photographed under an inverted microscope (Axio Vert. A1 ZEISS).

### 2.10. Statistical Tests

A two-tailed *T*-test was performed for all two sets of numerical data (treated and non-treated cells), and *p*-value ≤ 0.05 was considered statistically significant. Results are expressed as mean ± SD from experiments repeated at least three times. The statistical analysis was performed using the Statistical Package for the Social Sciences software (IBM SPSS Statistics for Macintosh, Version 27.0.).

## 3. Results

### 3.1. DPG Effect on Melanoma Cells Proliferation and Apoptosis

The anti-tumor effect of DPG was evaluated using a melanoma cell line (SK-MEL-28) carrying *BRAF* mutation. The cytotoxic effect of DPG observed in SK-MEL-28 cells was time- and dose-dependent, and for all further assays, the adopted half-maximal inhibitory concentration (IC_50_) was 36 mM for 24 h (Figure 1A). In addition, it was observed that cells presented nuclear morphological changes after 24 h of DPG exposure (Figure 1B). Moreover, the cytotoxic effect of DPG was evaluated in the non-tumoral HaCat cells by MTT assay, and the reduction in cell viability was observed only with high DPG concentrations (24–40 mM) for 48 h or lower DPG concentrations (8–40 mM) for 72 h (Appendix A).

Cell proliferation assay showed an anti-proliferative effect by DPG on melanoma cell line with significantly lower cell proliferation starting after 24 h of DPG exposure (*p*-value = 1.3 × 10^−7^) until 96 h later (*p*-value = 2.2 × 10^−14^) (Figure 1C). 

To investigate cellular apoptosis, DNA fragmentation was quantified by TUNEL assay. Thus, DPG treatment-induced apoptosis through DNA fragmentation (Figure 1D) was confirmed by a significant increase of TUNEL-positive cells (disrupted DNA stained by green fluorescence; intact cell nuclei visualized by DAPI (4′,6-Diamidine-2′-phenylindole dihydrochloride counterstaining) compared to untreated cells (*p*-value = 0.048) (Figure 1E). In this context, DPG also presented an apoptotic effect over the melanoma cell line.

In addition, *PARP-1*, *BCL-2*, and *BAX* expression levels were evaluated by qPCR in DPG-treated cells. Thus, it was found that treatment with DPG in melanoma cells decreased *PARP-1* (0.55 vs. 1.02 AUs, *p*-value = 0.001) and the anti-apoptotic gene, *BAX* (1.91 vs. 1.05 AUs, *p*-value = 0.09) expression (Figure 1F). Whereas the pro-apoptotic *BCL-2* mRNA level was significantly reduced in melanoma DPG exposed cells compared to control cells (0.51 vs. 1.07 AUs, *p*-value = 0.0018) (Figure 1F). These data confirm the induction of the intrinsic pathway of apoptosis by DPG.

Interestingly, no significant differences were observed in apoptosis-related genes in non-tumoral HaCat cells after 18 mM of DPG for 48 h (IC_50_) compared to control cells (*PARP-1*: 0.82 vs. 1.01. AUs, *p*-value = 0.16; *BAX*: 1.15 vs. 1.00 AUs, *p*-value = 0.70; *BCL-2*: 0.82 vs. 1.01 AUs, *p*-value = 0.26) (Appendix A).

Further, to investigate the effect of DPG on the migration ability of melanoma cells, SK-MEL-28 cells were treated with DPG for 24, 48, and 72 h, and a wound-healing motility assay was performed simultaneously. The results showed that cells exposed to DPG migrated significantly slower than DPG-free control cells (Figure 2A and Figure 3A), suggesting that DPG has an inhibitory effect on cell migration and invasion ability, characterizing it as an anti-tumor compound. Furthermore, the migration of SK-MEL-28 inhibited by DPG was evaluated by wound-healing assay. A significant decrease in cell migration was observed after 24 h (*p*-value = 1.0 × 10^−5^), 48 h (*p*-value = 4.0 × 10^−5^), and 72 h (*p*-value = 0.0003).

### 3.2. DPG Effect on Melanoma Stem-Like (SK-MEL-28) Cells

Next, it was investigated the phenotypic plasticity of cancer cells grown as melanospheres to elucidate the influence of DPG on some features of melanoma stem-like cells. Thus, it was observed that DPG promoted 100% of spheres breakdown and reduction in melanospheres formation compared to untreated cells, starting after 24 h of DPG-exposure (*p*-value = 0.008) (Figure 2B).

### 3.3. DPG Effect on miRs

A previously global analysis revealed that DPG could increase *miR-4443* and *miR-3620* expression levels (unpublished data), which are predicted to target *CD209* and *TNC* genes, respectively. Thus, the present study evaluated the expression of both miRs and their predicted target genes by qPCR using SK-MEL-28 cells exposed to DPG. In accordance with the global analysis, the mean mRNA expression level was significantly higher in DPG treated cells compared to control cells for *miR-4443* (1.77 AUs vs. 1.04 AUs; *p*-value = 0.02) (Figure 2C) and *miR-3620* (2.30 AUs vs. 1.00 AUs; *p*-value = 0.01) (Figure 2C). On the other hand, *CD209* (1.01 AUs vs. 0.54 AUs, *p*-value = 0.18) and *TNC* (1.00 AUs vs. 0.31 AUs; *p*-value = 2.38 × 10^−6^) expression levels were lower in DPG treated cells compared to controls (Figure 2D).

### 3.4. TPA-Induction of MMP-9 Activation and Migration of SK-MEL-28 Cells

TPA, a potent tumor promoter, has been used as a carcinogenetic inducer in vitro. The migration of SK-MEL-28 stimulated by TPA was evaluated by wound-healing and *MMP-9* mRNA expression. Wound-healing revealed a significant increase in the migration of SK-MEL-28 cells in the presence of TPA stimulation for 24 h (*p*-value = 0.002) and 48 h (*p*-value = 0.09) (Figure 3A,B) and by increased *MMP-9* expression level compared to control cells (3.56 vs. 1.08 AUs, *p*-value = 0.0004) 48 h after TPA-treatment (Figure 3C). In contrast, DPG inhibits the migratory effect of SK-MEL-28 in a time-dependent trend, in comparison with untreated control cells 24 h (*p*-value = 0.002), 48 h (*p*-value = 0.09), and 72 h (*p*-value = 6.3 × 10^−6^). Furthermore, wound healing showed that the migration of SK-MEL-28 cells stimulated by TPA was attenuated by adding DPG cells by the assay (48 h: *p*-value = 0.004; 72 h: *p*-value = 7.0 × 10^−4^). Further, the *MMP-9* expression level was inhibited by DPG in melanoma cells stimulated by TPA compared to only TPA-treated cells (0.99 vs. 3.56 AUs, *p*-value = 0.002) after 24 h of treatment. Our results suggested that DPG has an anti-migratory effect on SK-MEL-28 cells.

## 4. Discussion

A previous report demonstrated the potential role of DPG as an anti-tumor compound on glioblastoma cell lines [4]. In the present study, DPG was identified as a selective inhibitor of melanoma cells’ viability and melanoma stem cells, indicating that DPG is a promising anti-tumor drug for melanoma. The DPG anti-proliferative effect has been observed by cell proliferation and wound-healing assays [4] on glioblastoma cell lines. Here, we evaluated cell proliferation, apoptosis, and cell migration in a melanoma cell line and a non-tumoral (HaCat) cell line. Interestingly, the results on melanoma cells corroborated our previous findings [4]. In addition, HaCat showed a 40% cytotoxic effect for 48 h of DPG-exposure. Previously, HaCat showed the same toxicity percentage at the maximum TMZ concentration assessed (1000 µM) [17]. The chemotherapeutic regimens commonly used for the palliative treatment of malignant melanoma are intravenous administration of dacarbazine and oral administration of the alkylating agent (TMZ). TMZ has been well-tolerated, and it has an advantage in improving the quality of life of patients with metastatic melanoma [18].

Melanomas contain subsets of cancer stem-like cells with tumor-initiating capacity [19]. In this regard, it was also investigated the phenotypic plasticity of cancer cells grown as melanospheres to elucidate the influence of DPG on some features of melanoma stem-like cells. Thus, it was observed that DPG promoted a 100% reduction in melanospheres formation compared to untreated cells.

Given that apoptosis-inducing agents frequently signal through changes in the expression of BCL2 family proteins and possible alterations of BCL2 and BAX proteins, it was evaluated *BAX* (pro-apoptotic) and *BCL-2* (anti-apoptotic) after DPG-treatment (IC_50_ values). The *BCL-2* expression in melanoma cells was reduced dramatically after 48 h of treatment. Simultaneously with the downregulation of *BCL-2*, expression of *BAX* was increased upon treatment with DPG after 48 h. *BAX* expression increased by almost 50% in DPG-treated cells, indicating that melanoma is sensitive to DPG treatment. Otherwise, *PARP-1* mRNA expression levels presented as down-regulated in melanoma DPG-exposed cells. *PARP-1* interacts and modulates the function of several transcription factors, including NF-kB, NFAT nuclear factor (NFAT), E2F transcription factor 1 (E2F-1), and ETS transcription factor ELK1 (ELK-1) [20,21,22,23,24,25,26]. *PARP-1* is also involved in modulating endothelial cell adhesion molecule expression via its binding partner NF-kB [27,28,29] and as an inhibitor of MMPs [30]. Thus, our results indicate that, at least in part, both *PARP-1* and *MMP-9* are inhibited by DPG action, leading to abolishing melanoma cell migration.

Our study also revealed that DPG could increase *miR-4443* and *miR-3620* expression in melanoma cells, which are predicted to target the NF-kB post-transcriptional genes, *CD209* and *TNC*, respectively. Both target genes were observed as down-expressed in melanoma cells after DPG-exposure. CD209, also designated DC-SIGN, is a C-type lectin receptor present on both macrophages and dendritic cells, and it was preferentially located around tumors, colocalizing with Chemokine (C-C motif) ligand 18 (CCL18), regulating adhesion processes, such as DC trafficking, transient T-cell binding, and antigen capture [31,32]. TNC protein is a multifunctional matricellular glycoprotein, which is highly expressed in most melanoma cell lines, and it has been implicated in the progression of melanoma. A growing body of evidence has implicated the role of the *TNC* gene in the process of invasion and metastasis for melanoma [33]. Our results indicated that DPG up-regulates *miR-4443* and *miR-3620* and down-regulates their target genes, which might be involved in cerebral metastases formation; however, additional in vivo studies need to be performed to confirm the results.

The TPA, a mitogen and cancer promoter known as an inductor of tumorigenesis, is associated with increases in cell proliferation, epithelial to mesenchymal transition, and metastasis [34,35]. Also, TPA-induced cell migration in breast cancer MCF-7 cell line [36] and glioblastoma GBM8401 cell line [37]. Thus, our results using TPA-induced melanoma cell migration corroborated with previous studies. We observed an increase in *MMP-9* gene expression while stimulating the migration cells via wound healing assay by TPA on SK-MEL-28 cells.

In summary, our results suggested that DPG has an apoptotic, anti-proliferative, and anti-migratory effect on SK-MEL-28 cells. In addition, DPG was able to inhibit cancer stem-like cells that may cause cerebral tumor formation.

## Figures and Tables

**Figure 1 ijms-23-07251-f001:**
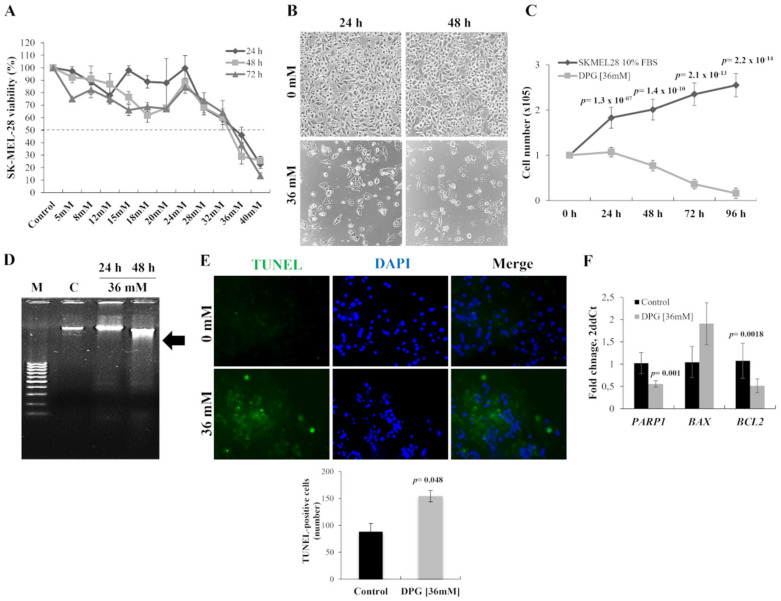
Dipotassium glycyrrhizinate (DPG) affects cell proliferation and apoptosis in a melanoma cell line. (**A**) DPG inhibits cell viability of SK-MEL-28 cells treated with different DPG concentrations for 24, 48, and 72 h by (4,5-dimethylthiazol-2-yl)-2,5-diphenyl tetrazolium bromide (MTT) assay and the half-maximal inhibitory concentration (IC_50_) was determined (36 mM for 24 h). (**B**) SK-MEL-28 nuclear morphological change was observed 24 h after DPG. Magnification = 400X. (**C**) DPG inhibits the proliferating rate of SK-MEL-28 in a time-dependent trend compared to untreated control cells. (**D**) SK-MEL-28 was incubated with 36 mM for 24 and 48 h, and the genomic DNA was isolated and analyzed on 1.5% agarose gel with ethidium bromide staining. M: DNA marker 100 base pairs; C: untreated control cells. (**E**) SK-MEL-28 cells treated with DPG (36 mM for 24 h) and terminal deoxynucleotidyl transferase (TdT) dUTP Nick-End labeling (TUNEL) assay was performed. Magnification = 400×. The quantitative estimation of TUNEL cells after DPG exposure was measured using ImageJ. The results show a significant increase in apoptotic cells (*p*-value = 0.048) after DPG. (**F**) DPG treatment in melanoma cells decreased Poly [ADP-ribose] Polymerase 1 (*PARP-1*) (*p*-value = 0.001) and the anti-apoptotic gene, BCL2 Associated X (*BAX*) (*p*-value = 0.09). Otherwise, the pro-apoptotic BCL2 Apoptosis Regulator (*BCL-2*) mRNA level significantly reduced melanoma DPG-exposure cells compared to control cells (*p*-value = 0.0018). For all assays, data represent means and standard deviations of representative experiments performed in triplicate. Statistics were performed in a two-tailed T-test with an alpha error of 0.05.

**Figure 2 ijms-23-07251-f002:**
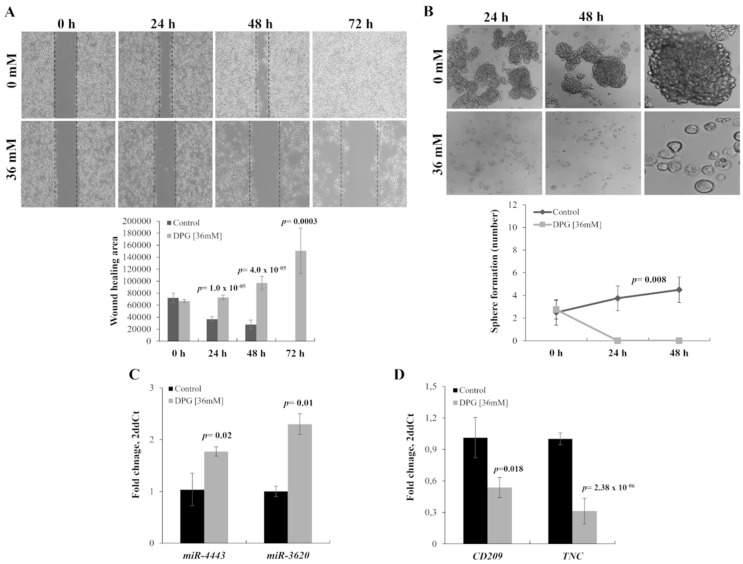
Dipotassium glycyrrhizinate (DPG) inhibits cell migration and cancer stem-like cells in a melanoma cell line. (**A**) SK-MEL-28 cells treated with DPG fill the wound area (the area between the two dotted lines) more slowly at 24, 48, and 72 h compared to untreated cells (*p*-value < 0.001). Graphics also shows the average and standard deviation of three independent experiments. (**B**) Loss of sphere-forming ability of SK-MEL-28 cells after DPG-exposure for 24 h (*p*-value = 0.008). At 48 h, DPG-treated cells were not available for analysis. The third image on the right shows just the enlargement of the images on the left. The graphic shows the average and standard deviation of three independent experiments. (**C**) DPG significantly increases *miR-4443* (*p*-value = 0.02) and *miR-3620* (*p*-value = 0.01) expression in SK-MEL-28 cell line. (**D**) DPG decreases cluster of differentiation 209 (*CD209*) and Tenascin (*TNC*) mRNA levels in SK-MEL-28 (*p*-value = 0.018 and *p*-value = 2.38 × 10^−6^, respectively). For all assays, data represent means and standard deviations of a representative experiment performed in triplicate. Statistics were performed in a two-tailed T-test with an alpha error of 0.05.

**Figure 3 ijms-23-07251-f003:**
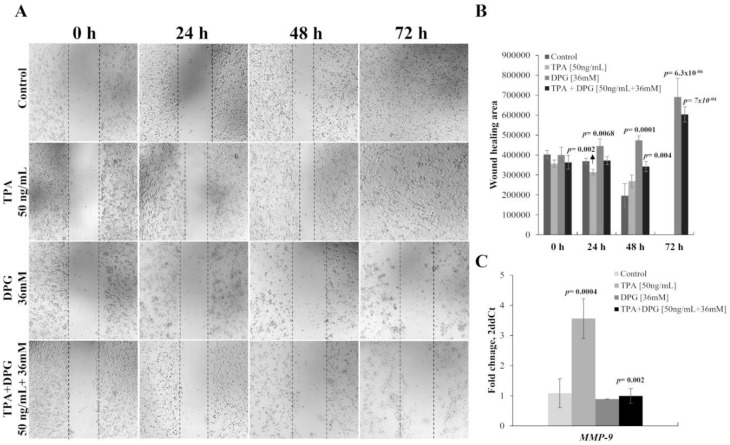
Melanoma cell migration stimulated by 12-O-tetradecanoylphorbol-13-acetate (TPA) was attenuated by adding dipotassium glycyrrhizinate (DPG) (**A**) TPA stimulates the migratory effect of SK-MEL-28 in a time-dependent trend, in comparison with untreated control cells. In contrast, DPG inhibits the migration of SK-MEL-28 in a time-dependent trend compared to untreated control cells. Finally, SK-MEL-28 cell migration stimulated by TPA was attenuated by adding DPG. (**B**) A significant increase in the migration of SK-MEL-28 cells was observed in the presence of TPA stimulation for 24 h (*p*-value = 0.002) and 48 h (*p*-value = 0.09). Furthermore, migration of SK-MEL-28 cells stimulated by TPA was attenuated by adding DPG cells by wound-healing assay (48 h: *p*-value = 0.004; 72 h: *p*-value = 7.0 × 10^−4^). (**C**) An increased Matrix Metalloproteinase 9 (*MMP-9*) mRNA expression level was observed compared to control cells SK-MEL-28 cells stimulated by TPA (*p*-value = 0.0004). Otherwise, decreased MMP-9 expression was observed in those DPG-treated SK-MEL-28 cells compared to untreated cells (*p*-value = 0.31) and in TPA plus DPG-exposure melanoma cells compared to control one (*p*-value = 0.39). In addition, the *MMP-9* expression level was inhibited by DPG in melanoma cells stimulated by TPA compared to only TPA-treated cells (*p*-value = 0.002) after 24 h of treatment. These results suggested that DPG has an anti-migratory effect on SK-MEL-28 cells. For all assays, data represent means and standard deviations of a representative experiment performed in triplicate. Statistics were performed in a two-tailed T-test with an alpha error of 0.05.

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
