# Peer review of "Dipotassium Glycyrrhizinate on Melanoma Cell Line: Inhibition of Cerebral Metastases Formation by Targeting NF-kB Genes-Mediating MicroRNA-4443 and MicroRNA-3620—Dipotassium Glycyrrhizinate Effect on Melanoma†"

_ijms, 2022, doi:10.3390/ijms23137251_

Round 1

Reviewer 1 Report

  1. Introduction part, please add on the sentence identified the linkage between NF-κB and BRAF mutation in melanoma.
  2. 2. “Results are expressed as mean±SD from experiments repeated at least three times. The statistical anal-ysis was performed by the Statistical Package for the Social Sciences software (IBM SPSS Statistics for Macintosh, Version 27.0.)” should be state the exactly statistical analysis which be used in the expression level as mean±SD.
  3. Figure 1B: SK-MEL-28 nuclear morphological change was observed 24h after DPG and Figure 1E, please add scale bar in Fig and power field in figure legend.
  4. Figure 2B: Loss of sphere-forming ability of SK-MEL-28 cells after DPG-exposure, please add scale bar
  5. Why the assays were performed until 72h?

Reviewer 2 Report

The manuscript by Bonafé et al. with a title “Dipotassium glycyrrhizinate on melanoma cell line: inhibition of cerebral metastases formation by targeting NF-kB genes-mediating microRNA-4443 and microRNA-3620” investigates effects of dipotassium glycyrrhizinate (DPG) on the melanoma cell line SK-MEL-28 and the non-tumor keratinocyte cell line HaCat. The manuscript is of interest and contains valuable data. However, there are some errors in the presentation and description of the, which require a major revision.

Major:

  1. 10 Which cell were used here? In the corresponding figure states that these are SK-MEL-28- please indicated it here as well.
  2. Figure 2B- Third image on the right shows just the enlargement of the images on the left. Please add it to the legend.
  3. Figure 2 C: I would delete this part and just cite the previous results as there are no other results / methods in the manuscript describing data generated using the glioblastoma cell line T98G. In addition, the Figure 2C is not easy to read and the information in the text would be sufficient.
  4. Please use the same names for the cell line SK-MEL-28 throughout the manuscript. I would always use the “SK-MEL-28”, eg. 2.4 “SKMEL” cells
  5. 4 Please add a reference investigating the effects of TPA and MMP-9 effects on melanoma.
  6. Figure 2A and Figure 3A- what kind of units are used in the chart? It would make sense to normalize the measured wound healing area to the control and to present it as a percentage of the control.
  7. Figure 3 A legend- “TPA stimulates the proliferation of SK-MEL-28” there are no data related to proliferation in this figure. Please change all parts that mention the proliferation to “migration” or “migratory effect”. Please delete the sentence in 2.4- “In contrast, DPG inhibits the proliferation rate…” as this data are not presented in this figure. This sentence could be rephrased.
  8. The title should be changed, as there is insufficient data in the manuscript for statements such as “inhibition of cerebral metastases formation” and “by targeting NF-kB genes-mediating microRnA-4443 and microRNA-3620”- there is only a measurement of expression changes of both microRNA after treatment with DPG, but not direct effects on melanoma cell behaviour by miR inhibition are investigated etc.
  9. The last paragraph of the discussion “previously, TPA, …” is unclear. Please make clear which results are described in the present manuscript and which statement refer to the literature /previous results.
  10. Effects on stem-like cell are not clear in the manuscript.

Minor:

The first sentence in the introduction could be also written in abstract, since it is not clear what kind of compound the dipotassium glycyrrhizinate is.

Figure 1 C: SKMEL28 can be deleted and only “10% FBS” used as a description

Round 2

Reviewer 2 Report

The authors have improved the manuscript according to reviewers’ comments. I have no more concerns.